# Toward Regenerative Sustainability: A Passive Design Comfort Assessment Method of Indoor Environment

**Kosara Kujundzic** [1,*], **Slavica Stamatovic Vuckovic** [2] **and Ana Radivojević** [3]

1    A TEAM Ltd., 85330 Kotor, Montenegro
2    Faculty of Architecture, University of Podgorica, 81000 Podgorica, Montenegro
3    Faculty of Architecture, University of Belgrade, 11000 Belgrade, Serbia
*    Correspondence: kosarak@gmail.com; Tel.: +382-69562088

**Abstract:** The fact that people spend a major part of their lifetime indoors, together with the lethal COVID-19 pandemic which caused people to spend even more time inside buildings, has drawn attention to the significance of achieving Agenda 2030 SD goal number three: good health and well-being, in reference to the indoor environment. The research subject is the health and well-being of building users explored through the sustainable (passive) design principles having an impact on the comfort and quality of the indoor environment. It is set within a regenerative sustainability framework encompassing the physiological, biophilic, psychological and social aspects of comfort. The Comfort Assessment Model's categories, to some extent, rely on the first author's doctoral thesis, with further modifications regarding the passive design criteria and indicators. A comparative analysis of the model with international sustainability certification (rating) systems has been performed, proving the significance of introducing more passive design comfort (health) related criteria into sustainability assessment models. In addition, a focus group of expert architects contributed to the research conclusions by responding to a questionnaire addressing the issues of sustainability, comfort and passive design, in terms of the health and well-being of building users, which confirmed the relevance of applied passive design measures for providing comfort indoors and fulfilling sustainable development goals.

**Keywords:** regenerative sustainability; sustainable architecture; passive design; humane design; biophilic design; comfort; indoor environment; international sustainability certification systems

## 1. Introduction

The fact that in modern society approximately 90% of a person's lifespan is spent indoors (dwelling, education, work, leisure, etc.), a considerable and growing number of user-oriented research demonstrates the relevance of the quality of the indoor environment for our health and well-being. Moreover, the COVID-19 lockdown situation and restrictions in allowing outside activities raised awareness of the health-related design aspects inside buildings. Therefore, a significant humane, sustainable design objective is the creation of comfortable and agreeable spaces, fostering users' health and well-being through a healthy and healing indoor environment.

The architectural discourse of sustainability has been constantly questioned and is changing towards a more comprehensive approach. However, sustainable architecture has always been related to health. It emerged as a concept during the 1990s, following the introduction of sustainable developmental approach within the "Brundtland report" in 1987 [1], as well as the UIA conference ''Declaration of Interdependence for a Sustainable future'' in 1993 [2]. Firstly, due to dealing with the energy and environmental crisis, the focus of sustainable design was on energy efficiency. Later, possibly as a consequence of performed energy saving measures, e.g., insufficient ventilation in order to save energy for heating, sick building syndrome occurred. Nowadays, the focus shifts from energy to

health, pollution to bio-diversity, social inclusion to visual impact [3]. However, design and building in accordance with nature, its processes, and the fundamental biological and psychological needs of human beings as an integral part of nature, as well as minimizing the building carbon footprint, remains the most significant principle of sustainable architecture, recognizing the health and well-being of the building users as high priority. Finally, the Agenda 2030 incorporated 'Good health and well-being' (SDG number three) among the 17 Sustainable Development Goals to be achieved by 2030, pointing out the relevance of the subject in terms of sustainability [4].

Within sustainable design development, we distinguish two main concepts: the dominantly engineering and technical approach (technological sustainability); and an ecological approach focused on the environment and living systems [5,6]. The research is set within the framework of an integrative (holistic) model of sustainable development whereby the ecological aspect prevails over the social and economic one, asserting the pre-eminence of nature [7]. Moreover, an ecological, holistic approach is the basis of regenerative development and design, which not only examines environmental impacts, but also the possibilities of regeneration on multiple levels in contrast to the energy efficient design (''green building'' or ''high performance building'') derived from the technological approach to sustainability, determined only through minimum or neutral environmental impact [8].

The regenerative approach represents a recent sustainability paradigm, considered as the next step in the evolution of sustainability [9]. Despite growing research focusing on the regenerative sustainability concept, design and assessment methodology remains underdeveloped in comparison to the prevailing traditional sustainable methods and models focused on topics of energy and minimized negative environmental impact. Therefore, new regenerative design assessment systems and methods should be introduced. In addition, the development of sustainability indicators should be oriented to the newly defined regenerative goals raising their benchmarks towards achieving a positive impact on humanity and the global environment [10].

Furthermore, despite the increasing number of sustainability assessment/rating systems, the focus maintains on 'green', energy efficient and physiological factors having an impact on the comfort/health of building users, e.g., adequate daylighting, acoustics, and air quality, while the biophilic and/or psychological/social aspects of comfort remain insufficiently present. In addition, passive design measures, an essential part of regenerative design methodology and highly relevant for health and well-being of building users, remain neglected and underrepresented. The paper addresses this research gap by carrying out a comprehensive literature review related to all the above mentioned regenerative design topics, demonstrating the interconnection and significance of all the comfort and health related quantitative and qualitative passive design aspects within the indoor environment. The main objective of the research is to enhance regenerative design assessment methodology by proposing a passive design comfort related model for the indoor environment encompassing physiological, biophilic and psychological/social aspects of comfort. The model reflects the salutogenic approach in architecture, oriented towards improving positive health outcomes and fostering various psychosomatic healing factors. Finally, introducing more bioclimatic, biophilic, salutogenic and comprehensively perceived comfort aspects within the sustainability assessment/rating systems, will contribute to achieving a relevant (regenerative) sustainability goal: good health and well-being.

## 2. Literature Review

The paper addressed literature on sustainability related to the following regenerative topics: salutogenesis, bioclimatic architecture, passive design, biophilic design, comfort, physical and mental health and well-being, all in reference to the indoor environment. The conceptual passive design comfort assessment model emerged from the theoretical background as a result of the comprehensive literature analysis. More specifically, thermal, visual, acoustic comfort, air quality, biophilic and psychological/social aspects have been considered, distinguished as relevant for improving the health and well-being of building'

users. This comfort related typology represents salutogenic discourse in architecture encompassing multiple, diverse, health-inducing factors contributing to achieving physical, mental and social well-being.

### 2.1. Regenerative Sustainability

The regenerative developmental approach refers to the harmonized co-evolving of all living beings, e.g., humans with nature, aimed at creatively enhancing vitality, viability, biodiversity and complexity. Moreover, the focus is on the life-supporting relationships between the built and natural environment [11]. It originates from the Regenesis Group, founded in 1995 by pioneering educators in the fields of permaculture and ecological design, Pamela Mang and Bill Reed, among others, whose early mission was to transform the development industry into one contributing rather than undermining the health of the planet [12]. The group carried out an educational, multidisciplinary program focused on regenerative practice, which led to the foundation of The Institute for Regenesis Practice [8,12]. Furthermore, the concept of regenerative development has relied on the General Systems Theory from the middle of the 20th century, introducing a new way of thinking based on change, growth and development, in opposition to the linear analytical approach dominant in science since the 17th century [13]. Moreover, the concept of regenerative development and design emerged from the idea of 'ecosystem design' brought by John Tillman Lyle in the 1980s, referring to the ecological and ecosystem's organization and order implemented in architecture [14]. Lyle explores the regenerative design further within the second book *Regenerative Design for Sustainable Development*, providing a comprehensive insight into regenerative principles and strategies, whereby the key architectural aspects incorporate the integration of natural elements/systems within passive design methodology (e.g., passive solar heating and cooling systems using plants for microclimate control, thermal storage, and the inducement of air movement within buildings) [15]. Similarly, regenerative buildings embrace natural light, air, and solar radiation as part of the indoor environment [15]. During the 1990s, an increased number of authors contributed to the redirection of ecological sustainability toward a more comprehensive, holistic, eco-systemic, regenerative approach, primarily focused on natural resource restoring rather than protection [6,16–19]. Today, regeneration represents the new sustainability's paradigm, incorporating the key terms of 'biophilia', 'health', and 'well-being', achieving the highest rank within the hierarchy of built environment consideration: from conventional, green, sustainable, restorative to regenerative [20–22]. Regenerative design is not about causing ''less damage'' to the environment, but rather about participating with the environment by using natural potentials and the health of ecological systems as a design base [23,24]. In contrast to the ''green design'', measurable by a growing number of standard/certification systems (LEED, BREEAM, CASBEE, etc.), regenerative design is less exact and tangible, without clearly defined quantifying parameters, but is more quality oriented, requiring more time and a systematic and complex approach [25]. Therefore, regenerative development and design implies different standards, closer but not limited to recently introduced standards (e.g., WELL-Building, and Living Building Challenge), which will synthetize technologically oriented "green" standards, and ecological, regenerative, and a more systematic and holistic approach to sustainability.

In addition, regenerative buildings should be not only less harmful to our health, but also significantly improve our well-being, in terms of physiological and psychological aspects, but, above all, through providing maximum comfort: adequate views, fresh air, daylight, pleasant temperature/humidity levels, connection to nature, etc. [26]. Moreover, the focus should be on the immaterial dimensions of sustainable design, i.e., the hallmarks become an holistic approach, salutogenesis, wholeness, synergy, symbiosis, and integration across the scales of local and global [27].

## 2.2. Salutogenesis

The term salutogenesis was coined in 1979 by Aaron Antonovsky, a professor of medical sociology, in the book *Health, Stress and Coping*, where he explores health-inducing factors, encompassing not only physical and biochemical parameters (essential for the survival), but also cultural, subcultural and individual responses to a constantly changing environment, crucial for achieving sociological and psychological homeostasis [28]. Antonovsky points out that the state of health exceeds a simple absence of disease, but is more complex and depends on numerous various factors. Furthermore, he establishes, i.e., conceptualizes, a salutogenic research model as a multidimensional continuum between the state of absolute health on one side, and the state of absolute illness (death) on the other, claiming that we move between those extremes without the possibility of achieving the state of absolute health. In addition, Antonovsky criticizes the World Health Organization's definition of health as a state of complete physical, mental and social well-being as impossibly abstract, philosophically utopian, deceiving and static [28]. Salutogenic, a multidisciplinary research model focused on the enhancement of healing factors applicable not only in medicine, but also in other natural and social sciences, represents a milestone in reference to dominantly present pathogenic model dealing with disease prevention and cure. More specifically, the salutogenic research subject is expanding curing and disease prevention factors to fostering and improving health and well-being aspects, thus moving towards the 'absolute health' side of the continuum. Therefore, in comparison to the reactive pathogenic approach (reaction to symptoms or indications of a disease), the proactive (providing conditions for physical, mental and social well-being) salutogenic research focuses on: health potential-'for health' (instead of 'against a disease'), and the creation of healing factors (rather than the elimination of risk factors) [29]. In the architectural realm, the salutogenic approach relates to a comprehensive, holistic research of design and building-related factors for improving the health and well-being of building users. Hence, salutogenesis is incorporated in the regenerative design methodology.

## 2.3. Bioclimatic Architecture, Passive Design

Environmental impact and the symbiotic relation to nature is of crucial importance to our evolution and survival on planet Earth. Furthermore, environmental preservation and the application of design principles in accordance with nature (the usage of renewable energy sources, air, sun heat and water movement, vegetation) represent the key influential factors for our state of health and well-being. These ecological concepts and methods are rooted in passive, bioclimatic design, whereby a building becomes an integral part of the environment, with regards to the topography and landscape, designed to mitigate, e.g., harmonize with day/night and seasonal changes in nature, aimed at achieving an energy efficient and comfortable indoor environment without using electricity [30–32].

Bioclimatic architecture represents an ecologically adequate, energy rational integration of the built environment within the natural environmental flow, aimed at achieving comfort in an overall sense [33]. Moreover, regenerative architectural design implements only $CO_2$ and electricity-free technologies, which implies the crucial significance of bioclimatic design methodology. In addition, it underlines achieving harmony with place [34], implying the relevance of building in accordance with the regional and local peculiarities, e.g., (micro) climate, topography, and autochthone materials, all relevant natural influential factors for bioclimatic architecture [35,36]. Passive design principles, the basis of bioclimatic design, represent an essential part of sustainable architecture providing comfort for building occupants while avoiding excessive energy consumption, expensive investments and complicated maintenance, which makes them economical and rational. These principles, introduced as archetype builders' reactions to natural environmental factors represented in vernacular, traditional, 'architecture without architects' have followed the development of humankind [37–39]. Despite their evolving over time, following the development of building techniques, materials and structural systems, the basic underlying concept remains the same: using and responding to the natural influential environmen-

tal factors (climate, topography, soil, vegetation) by saving energy and providing optimal conditions of comfort indoors. Nowadays, the implementation of passive design principles is constantly evolving owing to the development of the related software technology (e.g., orientation, form, openings) [40,41]. Passive design methodology contributes to energy efficiency, the preservation of natural resources, the reduction of environmental pollution, as well as to the improvement of the health of the building' users, hence to all three key aspects of sustainable development: ecological, social and economic, which is the reason behind its implementation worldwide [42–47].

In the paper, passive design measures are considered within the regenerative sustainability scope of humane design, concerned with the livability of all constituents of the global ecosystem, including plants and wildlife, focusing on human comfort (i.e., the comfort of building occupants) and enhancing the health and well-being of building' users [48–50]. It is noteworthy that passive design measures, as the simplest, cheapest, yet energy-efficient and environmentally favorable, should be implemented the first, while all other measures represent additional design and building methods, necessary only if the requirements exceed already applied passive design principles. More specifically, the first tier deals with basic building strategies (e.g., building orientation, insulation, and the use of exterior shading), followed by the second tier of passive or hybrid systems, and, lastly, mechanical equipment that could be incorporated, if needed, within an already passively optimized building design [51]. The proposed passive design comfort-related methodology encompasses physiological categories (thermal, visual, acoustic, air quality), as well as psychological/social and biophilic aspects, all relevant for the health and well-being of the building' users.

Building materials have a significant impact on the environment, all aspects of comfort and health directly or indirectly throughout the entire life cycle: extraction from the source (natural materials), the industrial process of creation (artificial materials); usage phase upon installation, and the destruction (recycling) phase upon expiring [52]. The research primarily focuses on the usage and maintenance phase when building materials most directly and intensively affect the indoor environmental quality and users' health and well-being. In addition, building materials are considered in reference to all introduced comfort related categories (thermal, visual, acoustic comfort, air quality, biophilic and psychological/social aspects). The crucial factors of the building materials' impact on users' health during this phase of direct exposure are the following: form and condition of a material (if material is loose, friable, containing volatile and/or radioactive elements); position within the building (contact with water, foodstuffs, internal/external, exposed/concealed); the means of degradation (mechanical, chemical action); the duration of periods of occupation and exposure; and maintenance cycles (if it may introduce toxic chemicals or increase dust resulting from maintenance) [53]. Natural materials are not only the most beneficial for the environment, but also for human health and well-being. More specifically, besides lower embodied energy and less processing required (i.e., less damage for the environment), they are generally lower in toxicity than man-made materials [48]. In addition, natural materials are hygroscopic (maintaining optimal humidity levels indoors), pleasant, 'inviting' to touch, carrying a message about time and place, thus fostering identity and genius loci, which all contributes to the enhancement of the state of health and well-being.

### 2.4. Biophilic Design

The term 'biophilia', meaning 'love of life' (in Greek: bios, philia) was introduced by Erich Fromm, a social psychologist, in 1964, referring to affinities of living beings to sustain life (from death threats) by mutual interaction and integration [54]. Twenty years later, a biologist named Edward Wilson used the term 'biophilia' in reference to human aspirations towards life and lifelike processes [55]. Bioclimatic design has emerged from the concept of biophilia since the beginning of the 21st century. Following the original definition, one of the pioneers of biophilic design, Stephen R. Kellert, argued that only the methods

inducing positive environmental impacts and the enhancement of people's physical and mental health, productivity and well-being, can be considered a biophilic design [56,57]. According to Zhong et al., the framework of biophilic design incorporates the three essential design approaches: (1) nature incorporation (naturally or artificially created natural elements providing multi-sensory experiences: water, air, daylight, plants, animals, landscape, weather, time and seasonal changes); (2) nature inspiration or biomimicry (evoke a sense of nature through the delicate placement of natural features such as forms, patterns, mechanisms, images, materials); and (3) nature interaction (the creation of nature-like settings fostering connections between various species within built environments) [57]. Despite the fact that biophilic design is still undeveloped in the architectural field, the benefits that the integration of natural elements within the indoor environment brings to the health and well-being of building users are certain. In terms of physiology, natural elements improve all aspects of comfort, e.g., plants and water features improve air quality (by tying polluters' particles, the creation of oxygen, pleasant aromas), efficiently mask noise, contribute to thermal comfort by evaporation, and act as shading devices preventing glare. Furthermore, natural elements such as potted indoor plants have a positive impact on cognitive and emotional functions, thus fostering health and well-being [58]. Finally, stress can be reduced and people are able to physically and psychologically heal more rapidly if connected with the natural (living) world [49]. To conclude, biophilic design principles, i.e., the integration of natural elements into the indoor environment, enhance conditions of comfort and foster users' health and well-being, and therefore directly contributes to achieving Sustainable Goal 3 of Agenda 2030.

*2.5. Comfort*

The discourse of comfort has been constantly changing throughout history, depending on social, economic and technological conditions and impacts. The beginning of the 21st century brings tendencies to move away from the perception of comfort as 'measurable physiological condition' toward qualifications underlining psychological, social and cultural aspects [59]. Overall, comfort could be defined in three ways: as a sensitized and satisfying relation between the human body and its immediate surroundings; as the enhancement of the immediate surroundings through the use of new technology and innovations in the field of architectural design and design in general; and as the mechanism of popular culture and the market's instrument, thus as one of the generators of consumer society [59].

Despite the evident complexity and diverse interpretation of the comfort phenomenon, the most documented is a conventional theory underlying physiological and sensory-accessible (perceptive) aspects of the surrounding, having direct impact on users, present in the international sustainability rating systems. However, even though these aspects remain crucial for understanding and studying comfort in the light of architecture, other intertwining qualitative and quantitative factors should be considered as well.

Various diseases have psychosomatic character. Therefore, commonly neglected psychological aspects of comfort have to be taken into consideration as inevitably attached to physiological factors. Furthermore, a connection to nature is of essence for health and well-being. Nature provides us with a connection to life cycles: birth, death and the restoring of life. It is dynamic, changing throughout the day and year, which provides an essential change to our senses, thus the perceptive diversity making our senses active and sharp and us awake and conscious. Furthermore, nature also enables us to achieve psychological satisfaction and emotional acceptance, which is important for our health. To conclude, connection to nature is relevant for our health and well-being and therefore an important design principle in humane design.

In other words, beside energy, ecological and socio-cultural factors, we should consider aesthetical, psychological and ambient aspects in the creation of the humane environment, which is not only healthy (not causing disease), but also inspiring, the source of delight and vitality, i.e., more salutogenic.

## 3. Materials and Methods

The starting point of the research is the first author's unpublished doctoral dissertation [60]. The thesis explores sustainable architecture/passive design assessment methodology in the sector of health tourism. Due to the crucial relevance for health and well-being, thus for both sectors: sustainable architecture and health tourism, comfort-related aspects (criteria, indicators, passive design measures) have been highly ranked within the passive design assessment model of health tourism facilities. This paper adopts some comfort related categories and arguments from the doctoral research, with further modifications of developed criteria and indicators. In addition, the assessment model is developed further in reference to the regenerative sustainability framework, especially in terms of the biophilic design aspects. Additionally, a major difference in comparison to the doctoral thesis is the comparative analysis with current sustainability rating systems carried out in the paper.

The research is based on a regenerative sustainability literature review encompassing the following key topics: salutogenesis, bioclimatic architecture, passive design, biophilic design and comfort. The passive design comfort assessment model derived from this theoretical background, comprising well-being and health-inducing aspects within the indoor environment, is divided into the following categories: thermal comfort, air quality, visual comfort, the biophilic aspects of comfort and the psychological/social aspects of comfort.

Furthermore, a comparative analysis with the sustainability rating systems have been conducted. Wider research scope involved Green Mark (Singapore), DGNB (Stuttgart, Germany), SNBS (Basel, Switzerland), and SBTool (Ottawa, Canada). However, the comfort assessment model is considered in reference to the following internationally popular and widely spread sustainability rating/certification systems (Table 1): U.S. Green Building Counsil's Leadership in Energy (LEED) [61], British Research Establishments' BREEAM [62], Japan Sustainable Building Consortium's CASBEE [63,64] International Living Future Institute's Living Building Challenge (LBC) [65], and International Well-building Institute's WELL-BUILDING (WELL) standard [66]. BREEAM and LEED are firstly introduced, then CASBEE, followed by LBC and finally, WELL as the most recent. The last editions referring to new construction and/or building are considered, regardless of the architectural typologies.

It is noteworthy that comfort and health related categories are intertwined and inseparable, underlining that health and well-being depend on comfort. Furthermore, the indoor environment category prevails in dealing with comfort issues, confirming the assumption of its high relevance to our state of health and well-being. Additionally, the youngest rating system-WELL incorporates the most comfort, thus health and well-being related categories/five out of ten, in contrast to all other systems covering only one category dominantly dealing with comfort.

Finally, the research was conducted through addressing a focus group of sixteen expert architects (university professors, architects-designers and scientific researchers) engaged in sustainable design (bioclimatic architecture) related to comfort, the indoor environment, and health and well-being aspects, for more than ten years. The experts have been consulted through the survey- questionnaire (Appendix A), in order to determine the relevance of the passive design measures and comfort-related categories, criteria and indicators within the passive design assessment model for users' health and well-being. In addition, the experts' responses ought to contribute to answering the main research question: the relevance of the introduction of more passive design measures as well as more comprehensive (biophilic, psychological and social) comfort related categories into sustainability assessment models of the indoor environment.

**Table 1.** Analyzed sustainability rating systems.

| Sustainability Rating Standard | LEED (USA) | BREEAM (UK) | CASBEE (Japan) | LBC (USA) | Well-Building (USA) |
|---|---|---|---|---|---|
| Year of First/Last edition | 1998/2019 | 1990/2018 | 2001/2014 | 2006/2019 | 2014/2020 |
| Structure/Main Sustainability categories | 1. Location and Transportation 2. Sustainable Sites 3. Water Efficiency 4. Energy and Atmosphere 5. Materials and Resources 6. Indoor Environmental Quality | 1. Management 2. Health and Well-being 3. Energy 4. Transport 5. Water 6. Materials 7. Waste 8. Land Use and Ecology 9. Pollution | 1. Q: Environmental Quality of Building *Q1-Indoor Environment Q2-Quality of Service Q3-Outdoor Environment (On-site)* 2. LR: Environmental Load Reduction of Building *LR1–Energy LR2–Resources & Materials LR3–Off site Environment* | 1. Air 2. Water 3. Nourishment 4. Light 5. Fitness 6. Comfort 7. Mind | 1. Air 2. Water 3. Nourishment 4. Light 5. Movement 6. Thermal comfort 7. Sound 8. Materials 9. Mind 10. Community |
| Categories dominantly dealing with comfort | Indoor Environmental Quality | Health and Well-being | Q1-Indoor Environment | Comfort | Air Light Thermal comfort Sound Mind |
| Categories partly dealing with comfort | Sustainable Sites Materials and Resources | Materials Land Use and Ecology | Q2–Quality of Service Q3–Outdoor Environment (On-site) | Air Light Mind | Movement Materials |

## 4. Passive Design Comfort Assessment Model

The Passive Design Comfort Assessment Method for the indoor environment encompasses six comfort-related categories [60]:

1. Thermal comfort
2. Air quality
3. Visual comfort
4. Acoustic comfort
5. Biophilic aspects of comfort
6. Psychological/Social aspects of comfort

The first four categories relate to physiological aspects of comfort, present within most of the sustainability rating systems. Physical aspects of comfort relate to providing: adequate indoor temperature relative to outside temperature; adequate relative humidity level and its impact on temperature; ample natural light and quality lighting without glare; adequate sound separation between buildings-from the outside and within a building, etc. [67] Biophilic aspects and psychological/social aspects are less tangible, thus more difficult for measuring, implying more qualitative and less quantifying criteria/indicators/measures.

### 4.1. Thermal Comfort

Heat (fire) is an archetype element essential for the maintenance of human vitality and life. In bioclimatic design, this element is primarily related to solar energy and thermal performances of a building, i.e., spatial layout, orientation, estimates of heat gain/loss through building envelope according to the applied materialization, and various other passive design measures (heating/cooling devices), all aimed at achieving optimal conditions of thermal comfort indoors. Life exists within a very small body temperature span of only a few degrees. Thermal comfort refers to an optimal sense of thermal agreeability (not too

hot, nor too cold) whereby a body thermal balance is achieved. Our health is dependent on conditions of thermal comfort. Overheating may lead to exhaustion, diminished working abilities and disagreeability, just as overcooling may. Thermal comfort depends on the six basic factors: activity, clothes, air temperature within indoor environment, mean radiation temperature, air velocity, and relative humidity [52].

With reference to temperature, the mode of heat transfer and how heat is produced, some warmth can be sleep inducing or even fatiguing, others energizing, relaxing or fostering relaxed well-being [3]. Heating by radiation and conduction warms up the body deeply, while convection heats up only the surface layers (skin and lungs). The conditions of thermal comfort can be fulfilled even if the air temperature is lower than optimal, if heat transfer is performed by radiation. Most agreeable conditions are those where the average radiation temperature (middle temperature of all radiant surfaces: walls, windows, floor, heating devices, furniture) is 2 °C higher than the air temperature [68]. In contrast to radiant heating, when heat transfer is by convection, higher temperatures are needed; the warmth is perceived as unpleasant, only felt at body surface, and the air quality is reduced (the spreading of dust particles, destroying negative ions), which causes fatigue and the deprivation of energy and vitality. The sun is the main heat source by radiation. The principles of bioclimatic design in this regard imply passive solar architecture, e.g., solar heat captivation and accumulation, in order to later use it for heating interior space during winter, autumn and spring, while at the same time providing protection from the excessive solar radiation- overheating during summer.

People have an inborn ability to adapt to changes in environmental conditions. For instance, when changing the environment, we immediately notice specific scents and noise, but get quickly adjusted to them as they fade into an ambient background. The same applies to thermal comfort, i.e., adjusting to changes in air temperature- adaptive approach to thermal comfort, which can be described as our proneness to adjust to change in air temperature, which has firstly caused the sense of discomfort, by striving to re-establish comfort [69].

In air-conditioned buildings, air temperature perceived as pleasant ranges from 22–25 °C, regardless of the outside air temperature. However, if a building is not air-conditioned but naturally ventilated, a comfortable air temperature indoors is dependent on the outside temperature and parallelly increases or decreases, which is the foundation of a variable, adaptive standard introduced by J. F. Nicol and M. A. Humphreys and defined by a formula: $T_c = 13.5 + 0.54 T_o$ ($T_c$- comfortable air temperature indoors, $T_o$- average monthly air temperature outside) [69].

The adaptive standard implies a wide span of thermally comfortable environments depending on the outside temperature in naturally ventilated buildings, which further implies achieving conditions of thermal comfort by applying passive design measures. In other words, applying passive design measures of solar architecture underlines our natural ability to adapt to thermal changes in the environment (air temperature), leading to the diminished usage of mechanical ventilation, which finally results in a healthier and more energy-efficient indoor environment.

In reference to the abovementioned influential factors, thermal comfort criteria and indicators within the proposed passive design comfort assessment model are presented in Table 2 as follows: (1) Form, Orientation, (2) Passive solar heating, (3) Passive cooling, (4) Thermal insulation, and (5) Windshield.

**Table 2.** Thermal Comfort Passive Design criteria and indicators.

| Criteria | Indicator | Passive Design Measures |
|---|---|---|
| 1. Form, orientation | 1.1 Building geometry (compactness, volume) | • Compact shape of a building (small building envelope in relation to the footprint area and volume) |
| | 1.2 Building Orientation | • South or South-east (12°–30° from south axis) |
| | 1.3 Rooms Orientation | • Daily used rooms oriented towards south, bigger glazed surfaces<br>• Night zones oriented towards north, smaller windows |
| 2. Passive solar heating | 2.1 Passive solar systems | • Massive walls, Trombe-Michel wall, water wall<br>• Thermal buffer zones (glazed balconies) |
| | 2.2 Materialization | • High Thermal Conduction and Capacity Materials (concrete, stone, solid brick, water)<br>• Dark colored materials<br>• Facades waterproofing (ventilated facades, coating or impregnation) |
| 3. Passive cooling | 3.1 Overheating Prevention | • Shading devices (canopies, pergolas, louvers)<br>• Vegetation (trees, green areas) |
| | 3.2 Passive cooling | • Solar (Thermal) Chimney<br>• Green and Water areas<br>• Natural Ventilation |
| 4. Thermal insulation | 4.1 Earth-sheltering | • North façade Earth-sheltered (if soil is dry) |
| | 4.2 Green roofs and facades | • Thick humus layer, dense vegetation |
| | 4.3 Materialization | • Low Heat Conductivity Coefficient of Thermo-insulating materials positioned on the outside edge of structural walls<br>• Facades waterproofing |
| 5. Windshield | 5.1 Natural barriers | • Vegetation (trees, bushes)<br>• Earth mounds |
| | 5.2 Artificial barriers | • Physical barriers (walls, buildings) |

Thermal Comfort within Sustainability Certification/Assessment Systems

Thermal comfort aspects are present in all analyzed sustainability rating systems, apart from the Living Building Challenge (Table 3). However, the criteria is mostly related only to active sustainable design measures. In addition, WELL standard is the only system recognizing relevance of providing thermal comfort in outdoor space as well as within the indoor environment.

**Table 3.** Thermal comfort aspects within Sustainability certification/assessment systems.

| Sustainability Certification System | Category/Subcategory | Criteria/Indicators/Passive Design Measures |
|---|---|---|
| LEED V4 | INDOOR ENVIRONMENTAL QUALITY (EQ)/ EQ Credit-Thermal comfort | • passive systems (night-time air, heat venting, or wind flow)-only for warehouses & distribution centres) |
| BREEAM | Health and Well-being/ 04 Thermal Comfort | • thermal modelling<br>• design for future thermal comfort<br>• thermal zoning and controls |
| CASBEE | Q1–Indoor Environment/ 2. Thermal Comfort | • room temperature control<br>• (room temperature, perimeter performance, zoned control)<br>• humidity control |

**Table 3.** *Cont.*

| Sustainability Certification System | Category/Subcategory | Criteria/Indicators/Passive Design Measures |
|---|---|---|
| LBC V4 | Not specified | Not specified |
| WELL-BUILDING | THERMAL COMFORT/ T01 Thermal performance T08 Enhanced operable windows T09 Outdoor thermal comfort | <ul><li>naturally conditioned regularly occupied spaces (outdoor temp.: min 10 °C, max 33.5 °C)</li><li>control of window operation</li><li>improve outdoor thermal comfort (vegetation-greenery, outdoor shading–e.g., canopies; reflectance of building materials and surfaces (e.g., sidewalks, rooftops); water features (e.g., ponds, fountains)-outdoor shading (at least 50% of pedestrian pathways and building entrances; at least 25% of parking spaces (if present); between 25% and 75% of all plazas, seating areas, exercise facilities with a contiguous area of less than 230 m and other outdoor areas of congregation.)</li><li>avoid excessive wind-winds exceed 5 m/s for more than 5% of hours in the year in seating areas or 10% of hours on paths and parking lots; not expected to exceed 15 m/s on paths, parking lots or seating areas for more than 0.05% of hours in the year.</li></ul> |

### 4.2. Air Quality

The comfort category of air quality is highly relevant for our health. Air components commonly having an impact on its quality are: oxygen and carbon-dioxide, carbon-monoxide, combustion products (tobacco smoke), dust particles, scents, building materials' emissions, ions and humidity.

Air pollution indoors is among the most serious health hazards. Sick building syndrome is related to inadequate air quality (strong draft, insufficient natural ventilation and air exchange, microbes and smell from air-conditioning units, dust particles, mites, and water within air humidifiers [70]. Kosoric implies five main groups of polluters: (1) biogenic particles (mold, bacteria); (2) combustion products (tobacco, gas appliances); (3) organic chemical from building materials (benzene, formaldehyde); (4) polluters occurred in the natural surrounding (radon), and (5) fibrose materials and particles (asbestos, glass wool, pollen) [71]. An increase in $CO_2$ levels of only 0.07% causes reduced alertness, lethargy, drowsiness and headaches [3]. Long term exposure to tobacco smoke may have serious consequences to health and cause cancer. Furthermore, dust particles are allergens which may contain numerous harmful components, i.e., pathogenic bacteria. Heating by convection spreads dust particles, the faster air convection, the more particles are spread out.

Natural ventilation and vegetation filtrate dust particles, decrease the concentration of CO, $CO_2$, tobacco smoke and other harmful air particles. Furthermore, vegetation enhances air quality by producing oxygen, the regulation of relative humidity and air temperature, absorbing polluters and creation of ions. Chrysanthemum morifolium, Dracaena deremensis and Gerbera jonesonii are most efficient in pollution absorbing, followed by Ficus benjamin, Hedera helix, Chamaedorea selfritzii and Spathiphyllum, while Schidapsus aureus successfully absorbs CO [3]. High levels of negative ions may inhibit spreading of pathogenic micro-organisms, which can be achieved by sunlight radiation and mobile air and water (fountains, cascades).

A significant factor of air quality are scents (aromas, fragrances). The sense of smell is bonded with the limbic system, responsible for especially intense emotional memory [72]. Pleasant aromas may decrease blood pressure, slow respiration and diminish pain-perception [73]. Scents are directly related to air temperature and humidity within indoor environment. Thus, at elevated temperature and humidity levels, smells are more perceived and, vice versa, lowered air temperatures diminish the intensity of scents. Simi-

larly, an increased quantity of fresh air brought indoors by natural ventilation reduces the intensity of smells. Furthermore, water absorbs odors and particles of air polluters. Due to evaporation, water decreases air temperature and if mobile enriches air with healthy negative ions.

Hygroscopicity is a property of porous earth-originated materials (clay, gypsum) and organic materials (wood, wool, plant fibers), which firstly absorb water during high humidity levels indoors, retain it in the pores until reaching low humidity levels and then release it into air, enabling the maintenance of optimal humidity levels, hence naturally improving air quality within indoor environment.

Breathing walls provide an exchange of gasses through the building envelope, which is relevant for regulation of harmful gasses and humidity levels. Gasses transfer is enabled in both directions: oxygen from the outside penetrates into building, while $CO_2$ exists outdoors. The diffusion of lighter gasses such as $CO_2$ is faster, while the movement of heavier molecules (e.g., polluters) occurs at a slower rate, i.e., a 20 cm thick brick wall's surface of 10 $m^2$ permits around 90l of oxygen per hour (under optimal pressure conditions), which fulfills oxygen needs of one person during the same time period [74].

Air quality comfort assessment criteria within the Passive design model are presented in Table 4 as follows: (1) Air cleaning, (2) Providing healthy air exchange rate, (3) EMF reduction, (4) Avoidance of geopathic zones, and (5) Materialization.

**Table 4.** Air quality passive design criteria, indicators and measures.

| Criteria | Indicator | Passive Design Measures |
|---|---|---|
| 1. Air Cleaning | 1.1 Vegetation | • Plants efficient as air cleaners (Chrysanthemum morifolium, Dracaena deremensis, etc.)<br>• Aromatic floral species |
| | 1.2 Water features | • Fountains, water cascades, open running water canals |
| 2. Providing healthy Air exchange rate | 2.1 Natural ventilation | • Operable windows<br>• Cross Ventilation |
| | 2.2 'Breathing' walls | • Vapor permeable walls |
| 3. EMF reduction | 3.1 Increased distance from EMF sources | • Frequently occupied rooms distant from electrical sub-stations or technical rooms with electrical equipment<br>• Furniture layout (a bed distant from a television or computer) |
| 4. Avoidance of geopathic zones | 4.1 Increased distance from sources of radon | • Elevated ground floors (if high concentration of radon occurs in the soil) |
| 5. Materialization | 5.1 Hygroscopic materials | • Natural materials: wood, earth, clay, cork |
| | 5.2 Non-toxic materials | • Avoidance of asbestos, benzene, formaldehyde, mineral wool<br>• Avoidance of radioactive materials<br>• Avoidance of excessive surface usage of metals |

Air Quality within Sustainability Certification/Assessment Systems

Air quality comfort related parameters within sustainability rating systems are presented in Table 5. The applied criteria mostly address issues of natural ventilation (operable windows), air quality monitoring and controlling measures ($CO_2$, chemical pollutants), and smoking prohibition.

**Table 5.** Air comfort within Sustainability certification/assessment systems.

| Sustainability Certification System | Category/Subcategory | Criteria/Indicators/Passive Design Measures |
|---|---|---|
| LEED V4 | INDOOR ENVIRONMENTAL QUALITY (EQ)/ EQ Prerequisite-Minimum indoor air quality performance EQ Credit-Enhanced indoor air quality strategies EQ Credit: Low-emitting materials EQ Credit-Indoor air quality assessment | • naturally ventilated spaces (procedure from ASHRAE Standard 62.1–2010 or a local equivalent) <br> • provide a direct exhaust airflow measurement device <br> • provide automatic indication devices on all natural ventilation openings intended to meet the minimum opening requirements. <br> • monitor carbon dioxide ($CO_2$) concentrations within each thermal zone. <br> • prohibit smoking inside the building <br> • reduce concentrations of chemical contaminants |
| BREEAM | Health and Well-being/ 02 Indoor air quality | • ventilation <br> • emission from construction products <br> • post-construction indoor air quality measurement |
| CASBEE | Q1–Indoor Environment/4. Air Quality | • source control (chemical pollutants; asbestos) <br> • ventilation (ventilation rate; natural ventilation performance; consideration for outside air take) <br> • operation plan ($CO_2$ monitoring; control of smoking) |
| LBC V4 | Health + Happiness/ I–09 Healthy Interior Environment | (ASHRAE 62–the standards for ventilation and indoor air quality) |
| WELL-BUILDING | AIR/ A01 Air quality A02 Smoke-free environment A03 Ventilation design A07 Operable windows | • prohibit smoking inside <br> • natural ventilation procedure in ASHRAE 62.1-2010 or more recent version <br> • provide operable windows-at least 75% of the spaces have operable window: for each floor, the openable window area is at least 4% the area of the occupied space. <br> • building entry design (entryway system composed, number of doors) and outdoor length) |

### 4.3. Visual Comfort

Visual comfort encompasses satisfying physical elements of comfort such as: qualitative illuminance (enabling comfortable eye adaptation and accommodation), contrasts (differences in brightness and/or color of objects) enabling distinguishing of objects; and fulfilling biological visual needs, i.e., the need for sunlight essential for hormonal system and synthetizing of vitamin D. On the contrary, glare (excessive brightness within sight scope) causes eye strain, which may lead to physical and psychological discomfort. A generally accepted design rule is to provide a sunlit floor surface- room depth equal to 1.5–2 times enlarged height of a window. Furthermore, the width of a window in relation to the wall mass around it affects daylighting and visual comfort, e.g., a smaller side window contrary to the darker background may cause glare and visual discomfort. Hence, the recommendation is to have more evenly and balanced distribution of sunlight within interior space, by bigger windows (size similar to the size of a wall), or by having windows on multiple facades [75]. Some researchers claim the minimum window size equal to 50% of the façade (with adequate shading devices to prevent overheating) [70]. It is noteworthy that windows in one wall simplify the quality of light, while windows in two or more walls enable more balanced light, reduce contrast (avoid over lit and dark spots) and replace silhouette with three-dimensional modelling [3]. The daylighting factor represents the daylighting level within a room measured as a percentage of that found outside on a horizontal surface [52]. The minimum required daylighting factor for kitchens is 2%, and for rest of the rooms 1,5%, while most people consider doubled minimum as comfortable

(2–5%). A vertical window size equal to 15% of the floor area provides minimum daylighting factor of 2% [52].

Color, as a property of light, affects the visual perception and atmosphere indoors, thus the visual comfort. Therefore, a more relaxed atmosphere can be induced by lower illuminance levels and warm light color, while higher illuminance and cold light color stimulate working atmosphere [76]. In addition, favorable interior colors depend on the climate conditions, i.e., cold light colors are adequate for hot climates while, in opposition to this, warm light colors are recommended for northern attitudes (cold climates). Therapy by color (chromatherapy), as an alternative, insufficiently science-based medical treatment, implies different modalities, from patients' exposure to colored lights, colored oil massage, color-focused visualizations, to consumption of colored food. Ocular light therapy, including light projected through colored filters into the eyes, is claimed to enhance mental, emotional and physical well-being and performance, which makes it efficient in treatment of stress/anxiety, insomnia/fatigue, headaches, and depression [73].

Visual comfort related to the passive design model incorporates the following criteria: (1) Daylighting/Windows, (2) Avoidance of glare, and (3) Visually stimulating design, presented with attached indicators and measures in Table 6.

**Table 6.** Visual comfort passive design criteria, indicators and measures.

| Criteria | Indicator | Passive Design Measures |
|---|---|---|
| 1. Daylighting/windows | 1.1 Windows size | • Minimum vertical windows size equal to 15% of the floor area (daylighting factor of 2%) |
| | 1.2 Windows layout | • Windows in two or more walls |
| 2. Avoidance of glare | 2.1 Shading devices | • canopies, pergolas, louvers, shutters<br>• Trees |
| | 2.2 Materialization | • Avoiding white, smooth surfaces (causing glare) in absence of shading devices, especially for floors |
| 3. Visually stimulating design | 3.1 Activity/color/illuminance ratio | • Cold colors (grey, green, blue) and higher illuminance levels for stimulating intellectual activities- offices, surgery rooms, libraries)<br>• Warm colors (yellow, orange, pink) and lower illuminance levels for stimulating activity (sport, recreation rooms) |

### Visual Comfort within Sustainability Certification/Assessment Systems

Visual comfort factors within Sustainability rating/assessment systems (Table 7) relate to adequate daylighting, providing views outside and the prevention of glare (anti-glare measures).

**Table 7.** Visual comfort within sustainability certification/assessment systems.

| Sustainability Certification System | Category/Subcategory | Criteria/Indicators/Passive Design Measures |
|---|---|---|
| LEED V4 | Indoor Environmental Quality (EQ)/ | • Daylight and Views–Daylight<br>• Daylight and Views–Views |
| BREEAM | Health and Well-being/ 04 Visual comfort | • Control of glare from sunlight<br>• Daylighting (building type dependent)<br>• View out<br>• Internal and external lighting levels, zoning and control |
| CASBEE | Q1–Indoor Environment/3. Lighting & Illumination | • Daylight (daylight factor; openings by orientation; daylight devices)<br>• Anti-glare Measures (daylight control; reflection control)<br>• Illuminance Level<br>• Lighting Controllability |
| LBC V4 | Health + Happiness/ I-09 Healthy Interior Environment | • Provide views outside and daylight for 75% of regularly occupied spaces |
| WELL-BUILDING | Light/ L01 Light Exposure L02 Visual Lighting Design L03 Circadian Lighting Design L05 Daylight Design Strategies L07 Visual Balance L09 Occupant Lighting Control | • At least 30% of the regularly occupied area is within 6 m horizontal distance of envelope glazing in each floor<br>• At least 70% of all seating in the spaces is within 5 m horizontal distance of envelope glazing.<br>• The envelope glazing area is no less than 7% of the regularly occupied floor area for each floor level.<br>• The floor plate is no more than 20 m between opposite walls that each have transparent envelope glazing, and there are no opaque obstructions higher than 1 m within a 6 m of horizontal distance to the transparent envelope glazing.<br>• Optimal distance from the façade (room depth) by 7 m, 70% of occupants<br>• 15–25% of the floor are equals to glazed envelope<br>• Visible light transmittance (VLT) is greater than 40%, (Manual and automatic shading) |

*4.4. Acoustic Comfort*

The most relevant aspects of acoustic comfort are providing adequate sound quality and avoidance of noise, which has physiological, psychological and aesthetic implications. Generally accepted comfortable sound level indoors is around 25 dBA, so that the close proximity of a road requires sound reduction of 45 dB (70 − 25 = 45 dB) [52]. This can be achieved by sound barriers (hard surfaces reflecting sound) and sound absorbers (soft materials, green absorbers: trees, bushes). Similar to thermal performances, façade doors and windows are less efficient soundproofing elements than solid walls, depending on the glazing and sealing quality (small holes around openings affect acoustic comfort indoors by not reducing sound levels). A sound level of 25 dBA is considered comfortable within the indoor environment, which implies the noise reduction of 45 dB (70 − 25 = 45 dB) if a building is next to a busy road [52]. This can be achieved with sound barriers and sound absorbers (soft materials, green absorbers: trees, bushes). Bigger absorbing surfaces are more efficient than smaller in sound reduction.

An energy efficient building, designed according to the passive design principles usually implies lower noise levels due to less mechanical ventilation, air-conditioning and heating, as well as adequate high quality windows blocking noise from the outside. Noise reduction indoors can be achieved by vegetation, while introducing moving water features (fountains, waterfalls) may significantly mask noise (the pleasant sound of water is closer to human ear, thus prevailing/'masking' the background noise). Acoustic comfort passive design criteria, indicators and measures are presented in Table 8.

**Table 8.** Acoustic comfort passive design criteria, indicators and measures.

| Criteria | Indicator | Passive Design Measures |
|---|---|---|
| 1. Noise screening | 1.1 Acoustic barriers | • buildings, walls, earth mounds, trees<br>• well-sealed doors and windows |
| | 1.2 Sound absorbers | • soft surfaces-materials<br>• big leaves vegetation |
| 2. Noise masking | 2.1 Vegetation | • flickering leaves vegetation |
| | 2.2 Water features | • mobile water features producing sounds (fountains, waterfalls) |
| 3.Sound proofing | 3.1 Materialization | • high quality soundproofing materials within partition walls and ceilings |

Acoustic Comfort within Sustainability Certification/Assessment Systems

Acoustic comfort related factors within Sustainability rating/assessment systems (Table 9) involves the following methods: soundproofing (sound insulation), sound absorbers and sound barriers.

**Table 9.** Acoustic comfort within sustainability certification/assessment systems.

| Sustainability Certification System | Category/Subcategory | Criteria/Indicators/Passive Design Measures |
|---|---|---|
| LEED V4 | Indoor Environmental Quality (EQ)/Minimum acoustic performance required | Prerequisite (only for schools):<br>• maximum HVAC background noise level of 40 dBA<br>• sound-absorptive finishes<br>• reverberation time requirements according to ANSI standard S12.60-2010<br>• acoustic treatment and other measures to minimize noise intrusion from exterior sources and control sound transmission between classrooms and other core learning spaces |
| BREEAM | Health and Well-being/Acoustic performance | • sound insulation<br>• indoor ambient noise levels<br>• reverberation time |
| CASBEE | Q1–Indoor Environment /Sound Environment | • noise levels 20–60 dB(A)<br>• soundproofing (openings, partition walls, floors/slabs)<br>• sound absorbers (materials) |
| LBC V4 | Health + Happiness/Healthy Interior Environment | • no specific acoustic criteria |
| WELL-BUILDING | Sound/<br>S01 Sound mapping S02 Maximum noise levels S03 Sound barriers S04 Reverberation time<br>S05 Sound reducing surfaces S06 Minimum background sound S07 Impact noise management | • label acoustic zones<br>• provide acoustic design plan<br>• limit background noise levels (level of dB(A))<br>• sound barriers according sound transmission class (STC); minimum Noise Isolation Class (NIC) or Weighted Difference Level (Dw)<br>• reverberation time, acoustical absorption<br>• performance of floor-ceiling materials |

*4.5. Biophilic Aspects of Comfort*

Human connection to nature is innate, since we are a part of nature, prone to its cycles of growth, change and transformation. Regardless of the technological development, our fundamental need to connect with the natural world through rudiment, complex sensory experiences, remains. The connection and synergy of humans and nature (ecology) is the research subject of ecopsychology, a science branch exploring connection of personal and global (planetary) health and well-being. A person cannot achieve the state of health

if the environment- nature is not healthy. Ecotherapy developed as an alternative medical approach aimed at connecting humans to natural systems on which our health and life depends. The notion of the healing potentials of nature exists from the beginnings of humankind. No healing environment can be achieved without connection to nature. Even artwork with natural scenes have a positive impact on health. In this regard, gardens are especially influential. Beside the relaxing effect, gardens affect emotion of satisfaction enabling easier wayfinding, i.e., orientation in space, which leads to a decrease of stress, especially important for patients and their families in healthcare facilities [77]. Gardens enable pleasant sensations and perceptions of nature: from inhaling fresh air rich in oxygen, feeling breeze on the skin, watching and smelling aromatic plants, to listening bird song, which all contributes to health and healing.

Nature is dynamic, constantly changing, with mobile, curved, complexed geometry full of transformative life forces it inspires people's inner mobility and energy. On the contrary, human thought and a straight line which does not belong to nature, dominates artificial spaces, being lifeless and dissimulative. Healing environments strive at a balance of intellectual order and energetic vitality, the harmony of artificial and natural. An artificial character of space can be mitigated by introducing nature indoors through: implementing natural materials (wood, brick, stone), retaining existing natural elements in the interior (rock, tree), or bringing natural features (plants, water) [78]. Natural materials have positive effect on our senses by the appearance with patina, carrying a message about material age, agreeable scent (e.g., wood), and pleasant, 'inviting' to touch texture. They connect us with the environment they derived from and support 'rootness' in life (awareness about local, relief, vegetation, climate), which elevates awakening of our senses, having positive impact on health. Life energies affecting our health involve aspects of connection to natural cycles: seasonal and daily rhythms of light, activity, sound and scents, evolving and decay [3]. Diverse sensory perception and connection to natural cycles and phenomena can be achieved by windows and daylighting (sense of sight, awareness of weather conditions, time of the day), as well as by natural ventilation (smell, sound, thermo-receptive skin sensors). The design principle of biomimicry -abstracting natural forms in curved shapes, architectural segments (niche, openings) and details (lighting features, handles, railings) interrupts spatial monotony and static character [78]. Furthermore, a 'natural feel' can be introduced by 'softening' window casted shadows through trees (leaves) in front of the window, or curved window edges [3].

Since ancient times, the human body was considered a result of a combination of the four fundamental cosmic elements: water, earth, fire and air. Water makes two thirds of our body, being responsible for all vital body functions. The element of earth refers to solid matter, essential for survival, as well as to the food we consume. Fire is present through body heat and energy, while the element of air relates to the respirational system, crucial for vitality. The issue of health/disease has referred to environmental impact on us and balance of those four elements. Furthermore, one of the biophilic design proponents, S.R. Kellert, introduces the fundamental natural elements (i.e., light, air, water, plants, animals, landscapes, weather, views, and fire) as biophilic design's attributes related to the direct experience of nature [79]. In order to fulfill optimal conditions of comfort, neither deprivation nor abundance of any element is recommended, but instead a harmonized balance. Healing environments contain the element of air through aerial flows (natural ventilation, draft), the element of fire (fire place, passive solar architecture), water (cascades, springs, fountains) and earth (potted plants indoors, clay) [78]. The scale of presence, quality and activity of elements in space affects the place character and attached attributes, e.g., dryness, lifelessness, fluidity, openness, introversion, warmth, etc.

Biophilic passive design criteria and attached indicators and design measures are presented in Table 10. The five main criteria incorporate: (1) Nature views, (2) Access to nature, (3) Introducing natural elements indoors, (4) Biomimicry-imitation of natural forms, and (5) Materialization.

**Table 10.** Biophilic passive design criteria, indicators and measures.

| Criteria | Indicator | Passive Design Measures |
|---|---|---|
| 1. Nature views | 1.1 Windows | • Nature views through windows (seasonally transformable landscapes, trees, plants)<br>• Softening window shadow edge (plants in front of windows (casting shadow through windows) |
| | 1.2 Artwork | • Nature motifs Art |
| 2. Access to nature | 2.1 Artificial elements | • Balconies, terraces, atriums |
| | 2.2 Natural elements | • courtyards with natural elements (flora, fauna)<br>• gardens with diverse vegetation |
| | 3.1 Retaining existing natural elements | • rock, tree |
| 3. Introducing natural elements indoors | 3.2 Natural features | • green areas<br>• water |
| | 3.3 Four fundamental elements (air, fire, water, earth) | • Air (natural ventilation)<br>• Fire (fire place, passive solar architecture)<br>• Water (fountains, waterfalls, water cascades, aquariums, open rainwater runoff canals)<br>• Earth (flower pots, clay elements, earth sheltering, vegetation) |
| 4. Biomimicry- imitation of natural forms | 4.1 Shapes | • Curved shapes supporting life energies and mobility<br>• Segments of space, openings, niche<br>• Architectural details: lighting features, door handles, railings |
| 5. Materialization | 5.1 Natural materials | • Wood, stone, brick, cork<br>• Local, autochthone materials |

Biophilic Aspects of Comfort within Sustainability Certification/Assessment Systems

Biophilic comfort related factors within sustainability rating/assessment systems (Table 11) involves the following methods: introduction of green areas (gardens, vegetated roofs, recreational space, potted plants, plant walls), and natural elements through implementation of natural materials, patterns, shapes, colours, images or sounds.

**Table 11.** Biophilic comfort related aspects within sustainability certification/assessment systems.

| Sustainability Certification System | Category/Subcategory | Criteria/Indicators/Passive Design Measures |
|---|---|---|
| LEED V4 | Sustainable Sites (SS)/ SS Credit: Open space SS Credit: Light pollution reduction | • create exterior open space that encourages interaction with the environment<br>• provide outdoor space greater than or equal to 30% of the total site area (including building footprint). A minimum of 25% of that outdoor space must be vegetated (turf grass does not count as vegetation) or have *overhead vegetated canopy*<br>• garden space with a diversity of vegetation types and species that provide opportunities for year-round visual interest<br>• vegetated roofs can be used toward the minimum 25% vegetation requirement<br>• increase night sky access, improve nighttime visibility<br>• places of respite and direct access to the natural environment (only for healthcare) |

**Table 11.** *Cont.*

| Sustainability Certification System | Category/Subcategory | Criteria/Indicators/Passive Design Measures |
|---|---|---|
| BREEAM | Health and Well-being/ 04 Visual comfort 07 Safe and healthy surroundings | • provide connection to nature by maximizing natural daylight and encouraging an external view out<br>• green recreational space brings an element of biophilia to a building by supporting human interaction with the natural environment |
| CASBEE | Q3 Outdoor Environment (On-site) | • preservation and creation of Biotope<br>I—Identification of local characteristics and biotope plan policy<br>II—conservation and restoration of biological resources<br>III—use of green space (securing the amount of greenery)<br>IV—quality of green space<br>V—Management and use of biological resources (Examples of efforts: *Provision of facilities for enjoying close contact with nature*)) |
| LBC V4 | Health + Happiness/ I-11 Access to nature Beauty/ I-19 Beauty + Biophilia | • biophilic design includes most of the requirements of the LBC 3.1 Biophilic Environment Imperative integrated with the requirements from the LBC 3.1 Beauty + Spirit Imperative |
| WELL-BUILDING | Mind/ M02 Nature and Place M09 Enhanced Access to Nature | • incorporate natural elements into buildings: plants (e.g., potted plants, plant walls); water (e.g., fountain); nature views or representational (e.g., photographs), natural materials, patterns, shapes, colours, images or sounds<br>• incorporating other key aesthetic elements (local culture, art, etc.) |

*4.6. Psychological/Social Aspects of Comfort*

Psychological and social aspects affect our physical and mental health and well-being. Around half (35–70%) of curation is a result of the 'placebo effect', even 70% when a doctor believes in therapy [3]. Furthermore, most of the illnesses have psycho-somatic character. However, these aspects are neglected and insufficiently present within sustainability assessment/rating systems. The regenerative sustainable design tendencies embrace holistic approach to health and sustainability with interdisciplinary approach encompassing psychological/social aspects. Within the passive design assessment model, seven criteria are distinguished (Table 12): (1) Constant and controlled change, (2) Visual aspects, (3) Bonding building with place and time, (4) Adaptability and flexibility, (5) Safety and accessibility, (6) Social support, and (7) Materialization.

**Table 12.** Psychological/social comfort passive design criteria, indicators and measures.

| Criteria | Indicator | (c) Passive Design Measures |
|---|---|---|
| 1. Constant and controlled change | 1.1 Physiological | • Visual, thermal, acoustic, aerial |
| 2. Visual aspects | 2.1 Views | • Windows, nature view<br>• Artwork (works of art with natural motifs, positive moods and emotions-paintings, sculptures, reliefs, scenography) |
| | 2.2 Color | • Cold colors (grey, green, blue) for stimulating intellectual activities-offices, surgery rooms, libraries), warm climates<br>• Warm colors (yellow, orange, pink) for stimulating activity (sport, recreation rooms), cold climates |
| | 2.3 Form | • Human scale places<br>• Soft, curved shapes for supporting life energies and mobility (physical activity)<br>• Hard, rectangular shapes for stimulating intellectual processes |

**Table 12.** *Cont.*

| Criteria | Indicator | (c) Passive Design Measures |
|---|---|---|
| 3. Bonding building with place and time | 3.1 'Rooting' building into the ground | • Earth-sheltering<br>• Vegetation attached to building |
| | 3.2 Materialization | • Local, autochthone, natural materials |
| 4. Adaptability and flexibility | 4.1 Physical | • Space adaptable in layout and size |
| | 4.2 Functional | • Space adaptable in use |
| 5. Safety and Accessibility | 5.1 Physical | • Rooms accessible for all users (disabled) |
| | 5.2 Social (Psychological) | • wayfinding, human scale, hospitable, familiar, 'domestic' atmosphere |
| 6. Social Support | 6.1 Physical | • Rooms adaptable to different size groups<br>• Furniture layout (round tables instead of rectangular, seating 'in circle' instead of in rows) |
| 7. Materialization | 7.1 Tactility of materials | • Warm, natural materials (wood, brick)<br>• Materials pleasant to walk on (sand, gravel, pebbles) |

The need for change within the environment is a basic, psychological need. A deprivation of change leads to the inactivity of senses, decreased levels of focus, attention, sensitivity, perception, which finally results in lethargy and negative moods. The changes can be visual (lighting levels, color), functional (multi-use spaces), organizational (layout of walls, divisions, furniture and equipment), thermal (thermal variability), acoustic (diversity of sounds), and aerial- air exchange indoors (ventilation). Some researches of healthcare facilities proved the importance of diversity of space (visually available versus visually closed) and multisensory 'retreats' within a building for the emotional and cognitive functions which may affect the immune system [73]. Passive design measures support thermal and visual variability of the indoor environment, regarding the different room orientation and layout, as well as functional zones according to the sun. However, overstimulation is not preferable, because of inducing strain, fatigue and stress. In conclusion, moderate, constant, controlled change within all perceptive comfort aspects is favorable.

Color is a medium prone to subjective interpretations, personal affinity and perception. The psychological effects of color may change the atmosphere and perception of space. Regarding the psychological aspect, within cold climates, warm colors are favorable, in opposition to warm climates where the preference is cold colors. Warm colors inspire activity, liveliness, and extroversion, while cold colors initiate introversion, contemplation and intellectual processes.

A view is a basic visual, psychological need, implying daylight and windows. View enables connection to the outside world and nature, provides information about the time of the day, weather conditions, surrounding, thus helping us orienting in space and time. View on 'psychologically adequate' artwork showing themes from nature (landscapes, flowers, gardens) and well as figural art showing emotionally positive gestures and facial expressions, may reduce stress and enhance health outcomes such as pain relief [77].

Topography is an important factor of building 'rooting'. Instead of adjusting terrain to the house, we should homogenize the natural and built environment by harmonizing the house to the land configuration. In sloping terrain, a building adapts by earth-sheltering. On flat land, 'anchoring' is achieved by bushes attached to a building, or expanding of the ground floor where connected to the ground [3]. Autochthone materials derived from the surrounding carry the trace of time, creating a feeling of building belonging to the place of origins, with regard to being transformed by the natural influential factors. Hence, stone and brick change shape and gain patina over time, wood changes color from natural toward grey, old wall paints fade out. The use of local materials enhances the feeling of identity (genius loci), simplifying orientation in time and space, in contrast to universal, industrial materials almost inert toward the environment, independent from the place of

origins. Seasonally transforming plants, colors washing off over time and other aspects respond to ephemeral and attach a building to time and life. Another psychologically significant factor of building materials is tactility. Frampton points out that tactility is a significant feature of form and space, because of the possible perception only through a direct experience, which cannot be reduced to a simple information or evocation of simulacrum replacing absence [80]. Warm materials, pleasant to touch, create an agreeable sensation and positively psychologically affect building users, i.e., inspire stay within a space and interaction by creating a wide span of sensations, e.g., a feeling of warmth/cold, hard/soft and smooth/rough.

Spatial adaptability and flexibility, in terms of simply adjusting to changes of the occupants' needs and usage mode, represents a significant factor of psychological/social aspects of comfort. In reference to this, rooms should be physically adaptable in layout and size, i.e., adaptable to different size groups, which enhances social support, as well as furniture-seating layout: seating 'in circles' instead of 'in rows' suggesting equality, round tables replacing 'hierarchy implying' rectangular ones (the highest significance is of a speaker in front, the lowest of those seating in the last row). Moreover, seating 'shoulder to shoulder' along walls restrains social interaction, while placing tables and chairs in the middle of a room increases interaction [77].

Humane design refers to safe and accessible indoor environment for all building users. Regarding the physical aspect, all rooms should be adjusted to most sensitive category of users (elderly and disabled) by the avoidance of physical barriers and slippery floors. In terms of the psychological/social safety and accessibility design priorities are: human scale, wayfinding (visually distinctive accents: gardens, windows, atriums) and achieving hospitable, familiar, 'domestic' atmosphere. In addition, the form and size of the interior space have psychological implications for health. Too high and immense a space can have an intimidating effect on users, while human scale rooms inspire pleasant and welcoming atmosphere. Furthermore, shapes can induce psychological reactions, e.g., hard, angled and rectangular forms stimulate intellectual clarity followed by feelings of ascetic calmness to repulsive inaccessibility, while soft and rounded strives at sensuality varying from welcoming to oppressively enwrapping [3].

Passive design psychological/social comfort related criteria, indicators and measures are presented in Table 12.

Psychological/Social Aspects of Comfort within Sustainability Assessment Systems

Psychological/social aspects of comfort within sustainability rating/assessment systems (Table 13) involve mostly aspects of physical safety (barrier-free, inclusive methods), and outdoor spaces adjusted to social activities and/or restorative interaction within natural setting.

**Table 13.** Psychological/social comfort related aspects within sustainability certification/ assessment systems.

| Sustainability Certification System | Category/Subcategory | Criteria/Indicators/Passive Design Measures |
|---|---|---|
| LEED V4 | Sustainable Sites (SS)/ Prerequisite: Environmental site Assessment Required, Credit: Open space, Credits: Places of respite and Direct exterior access (only for healthcare) | • protect the health of vulnerable populations (schools, healthcare)<br>• development density and community connectivity<br>• create exterior open space that encourages social interaction, passive recreation, and physical activities<br>• outdoor space must be physically accessible (accommodate outdoor social and physical activities; garden space with a diversity of vegetation types)<br>• a garden space dedicated to community gardens or urban food production<br><br>Only for healthcare buildings:<br>• place of respite is accessible from within the building or located within 60 m of a building entrance or access point, may not be within 7.6 m. of a smoking area and is open to fresh air, the sky, and the natural elements<br>• shade or indirect sun-with at least one seating space per 18.5 square meters of each respite area, with one wheelchair space per five seating spaces<br>• horticulture therapy and other specific clinical or special-use gardens unavailable to all building occupants may account for no more than 50% of the required area<br>• universal-access natural trails that are available to visitors, staff, or patients may account for no more than 30% of the required area, provided the trailhead is within 60 m of a building entrance |
| BREEAM | Health and Well-being/ 06 Security 07 Safe and healthy surroundings | security of site and building (Private space-provision of outdoor space which gives privacy and a sense of wellbeing)<br>• safe access and movement around the site and outdoor space<br>• facilitate the activities that can have physical, mental and social benefits |
| CASBEE | Q2–1. Service Ability 1. 1 Functionality & Usability 1. 2 Amenity | • barrier-free plan<br>• perceived spaciousness and access to view (ceiling height, good view as psychologically comfortable)<br>• space for refreshment (focus on office building)<br>• décor planning (natural and ecological materials) |
| LBC V4 | PLACE/ 4 Human-Scaled Living EQUITY/ 17 Universal Access 18 Inclusion | • walkable, pedestrian-oriented communities<br>• provide places for occupants to gather and connect with the community; sufficient secure, weather-protected storage for showers and lockers, to encourage biking; electric vehicle charging stations<br>• infrastructure and features (e.g., plazas, seating or park space, street furniture, public art, gardens, etc.) equally accessible to all, regardless of background, age and socioeconomic class)<br>• safeguard access for those with physical disabilities |
| WELL-BUILDING | MOVEMENT/ V04 Facilities for active occupants V05 Site planning and selection V06 Physical activity opportunities MIND/ M07 Restorative spaces COMMUNITY/ C13 Accessibility and universal design | • promote and encouraging movement through site and variety of positions throughout the day-ergonomic design solutions<br>• encourage healthy behaviours, such as stairclimbing<br>• enable inclusive entrance, easy access to all spaces and amenities and minimize risk of injury, confusion or discomfort (e.g., lighting or clear sightlines to increase feelings of security)<br>• providing restorative spaces (through incorporation of nature and natural elements) |

## 5. Results and Discussion

A bioclimatic (passive) design comfort assessment model is defined as a result of the literature review. Within the proposed comfort assessment model, comfort-related factors are considered not only through the physiological aspects of comfort (thermal comfort, air quality, visual and acoustic comfort), but also through the less quantified, but more qualitatively determined biophilic and psychological/social aspects. Bioclimatic design emerged from archetype builders' reactions to natural influential factors (sun, climate, landscape, topography). Passive design measures represent main bioclimatic design methodology,

highly beneficial in terms of environmental impact and the enhancement of building users'
comfort within indoor environment. These measures are inevitable in sustainable design
vocabulary, implemented first, while active measures ought to be optional, 'if needed',
only if the requirements are not entirely met by application of passive design measures.
Passive architectural methods are ecological, $CO_2$-free, incorporate renewable energy re-
sources, i.e., they improve energy efficiency, and do not require high investments, hence
they contribute to the economic aspect of sustainability. Moreover, those methods are fa-
vorable in terms of health, well-being and the comfort of building users, due to encompass-
ing people's innate connection to complex and constantly changing natural environment.
These biological, physiological and psychological relations and interdependences of hu-
mans and other living beings, essential for health, are emphasized in biophilic design, by
fostering physical and psychological connections of people and nature, i.e., of the built and
natural environment.

　　Furthermore, a comparative analysis of internationally popular and widely spread
sustainability assessment/rating systems: LEED, BREEAM, CASBEE, LBC and WELL ac-
cording to the six proposed comfort-related categories is performed confirming insuffi-
cient presence of comfort-related aspects, especially within the domain of passive design,
all in reference to the indoor environment. Firstly, dominantly comfort-related sustain-
ability categories are reduced to only one in all rating systems, except of WELL (five).
Moreover, WELL is particularly oriented towards covering individual needs of building
occupants through addressing aspects of their physical and psychological health and well-
being. However, the comfort aspects within domain of architectural and passive design
are barely considered. Instead, monitoring, management and maintenance categories pre-
vail. In addition, the physical (physiological) aspects of comfort remain dominant in all
analyzed systems in comparison to biophilic and psychological/social factors. The pro-
posed passive design comfort assessment criteria/indicators/measures overall presence in
reference to comfort category is compared to sustainability rating systems (Table 14). The
tendency of introducing more qualitative and intangible assessment factors is recognized
within more recent rating systems, e.g., biophilia/qualitative in addition to biophilia/
quantitative in the first edition of the youngest system-WELL/V1, introduced in 2016. The
comparative analysis confirmed the insufficient presence of the relevant regenerative de-
sign topics (bioclimatic, passive, biophilic design, and comfort-related aspects) in refer-
ence to the indoor environment within the recent sustainability rating systems, therefore,
proved the relevance of the research topic.

**Table 14.** The presence of passive design comfort assessment model categories/
criteria/indicators/measures in sustainability rating systems.

| | Comfort Category of Passive Design Assessment Model | LEED (USA) | BREEAM (UK) | CASBEE (Japan) | LBC (USA) | Well Building (USA) |
|---|---|---|---|---|---|---|
| 1 | Thermal comfort | − | − | − | − | −+ |
| 2 | Air quality | +− | +− | +− | −+ | +− |
| 3 | Visual comfort | + | + | + | + | + |
| 4 | Acoustic comfort | + | + | + | − | |
| 5 | Biophilic aspects of comfort | +− | −+ | −+ | −+ | + |
| 6 | Psychological/Social Aspects of Comfort | −+ | −+ | −+ | −+ | −+ |

+: more than 80% of criteria/indicators covered. +−: majority of criteria/indicators covered. −+: minority of
criteria/indicators covered. −: no criteria/indicators covered.

　　Finally, a focus group of 16 expert architects, university professors (eight), architects-
designers (six), and scientific researchers (two), all involved in diverse but intertwined
sustainable design categories (bioclimatic architecture, energy efficiency, health and age-

ing, place making, and participatory design) for more than 10 years have reviewed the proposed passive design comfort assessment model through a questionnaire, by rating level of relevance for subsequently: Sustainable Design (regarding fulfilling SDG number three- Good health and Well-being); Passive Design Measures (in reference to health of building users), and Comfort Categories/Criteria/Indicators/Design measures with regards to health and well-being of building users. The summarized results presented in the Appendix A confirm the high level of relevance of implementing passive design measures in all comfort-related categories within indoor environment, in order to enhance health and well-being of building users. More specifically, indoor environmental quality, passive design measures, and all six comfort-related categories within the passive design model are in overall evaluated as 'of highest level of relevance' for building users' health and well-being. The survey results provided a valuable insight into the significance of currently underrepresented comfort-related categories and passive design measures, defined within the regenerative, i.e., salutogenic framework. In reference to this, a further developmental approach would be setting a hierarchy in criteria/indicator/passive measure's relevance in enhancing comfort, thus health and well-being of building users.

## 6. Conclusions

A considerable amount of research has confirmed that we spend the majority of our lifespan indoors. This fact alone demonstrates the significance of indoor environmental quality, i.e., architecture, for our health and well-being, hence to the quality of our life. In addition, within the architectural realm, the issue of health is intertwined with the notion of the comfort of building occupants, constantly representing high priority in sustainable (humane) design.

Furthermore, sustainability discourse in architecture is recently shifting towards a more comprehensive and systematic framework, while a regenerative sustainability approach represents the new design paradigm, moving away from a linear, analytical way of thinking towards the considerations of change, growth, transformation and the co-evolving of living beings, e.g., people within the natural world. The key terms within the regenerative design field become: biophilia, salutogenesis, health, well-being, synergy, as well as bioclimatic design, prioritized in the architectural domain. Therefore, regenerative design requires new, more holistic methodology, involving qualitative and intangible categories enhancing health and well-being of building users. This methodology, encompassing various biophilic, psychological/social, and passive design comfort related factors, is currently underrepresented, thus undervalued within the international sustainability rating systems (LEED, BREEAM, CASBEE, LBC, and WELL). In this regard, the paper addresses passive design comfort related methodology set within the salutogenic research framework exploring various aspects contributing to physical, mental and social well-being of building users. The comfort assessment model emerged from the comprehensive literature review of regenerative sustainability topics (salutogenesis, bioclimatic architecture, passive design, and biophilic design). In addition, a comparative analysis with the sustainability rating systems confirmed insufficient presence of relevant, health-inducing regenerative design methods in reference to the indoor environment. Moreover, the responses of the focus group of experts in sustainable design have demonstrated a high level of relevance of all passive design comfort related categories/criteria/indicators presented in the assessment model in regard to health and well-being of building users.

Limitations of the conducted research are recognized in its generic character. More specifically, comfort assessment categories, indicators and passive measures are considered regardless of the architectural typologies. Furthermore, diverse requirements depending on building types may confirm the necessity of introducing new and more typology-specific comfort-related categories, indicators and measures. However, the main orientation of the research is towards raising awareness of the significance of introducing more passive design measures, as well as diverse, qualitative comfort-related aspects (e.g., biophilic and psychological/social aspects) into sustainable design and sustainability assess-

ment systems/models. Therefore, its most relevant contribution is enhancing regenerative design/assessment methodology with a passive design comfort-related model addressing various and diverse, but intertwined and inseparable, more qualitative, less quantified, thermal, visual, acoustic, biophilic and psychological/social aspects of comfort, all highly relevant for enhancing building users' health and physical, mental and social well-being, therefore leading to achieving Agenda 2030′s sustainability goals, especially goal number three: good health and well-being.

**Author Contributions:** Definition of methodology and frameworks, writing and review, K.K.; literature review, comparative analysis, writing, S.S.V.; review, editing. A.R. All authors have read and agreed to the published version of the manuscript.

**Funding:** This research received no external funding.

**Institutional Review Board Statement:** Not applicable.

**Informed Consent Statement:** Not applicable.

**Data Availability Statement:** Not applicable.

**Acknowledgments:** The authors would like to express their gratitude to all professionals who participated in the survey and contributed to research results.

**Conflicts of Interest:** The authors declare no conflict of interest.

## Appendix A

### Questionnaire on Passive Design Comfort Assessment Methodology

\* Summerized survey results

**Introductory questions**

1. What sector are you engaged in? (e.g., public/private/higher education/non-profit)

Answer:

| Sector of Engaging of Participants: | | | | | Total Number |
|---|---|---|---|---|---|
| Higher Education | Private | Private & Higher Education | Private and Non-Profit | Non-Profit | |
| 8 | 4 | 2 | 1 | 1 | 16 |

2. What is your occupation/area of expertise? (e.g., Architect-designer/sustainable design; University professor/ bioclimatic architecture, etc.)

Answer:

| Occupation/Area of Expertise of Participants: | | |
|---|---|---|
| University Professors * | Architects–Designer ** | Scientific Researchers *** |
| 8 | 6 | 2 |

\* Bioclimatic architecture/Energy efficiency;/Sustainable architecture/Health and ageing, place making, participatory design; \*\* Architect–practice/Environmental design and planning/Energy auditor; \*\*\* Bioclimatic architecture/Sustainable design.

3. How many years of professional experience related to sustainability do you have?

Answer: Average-18 years (min 10, max 40)

4. Which sustainability category are you mostly involved in? (multiple answer question)

| Sustainability Category (According to the LEED Rating System) | Answer (x) |
|---|---|
| 1. Location and transportation | / |
| 2. Sustainable sites | 28% |
| 3. Water efficiency | 3% |
| 4. Energy and atmosphere | 19% |
| 5. Materials and resources | 22% |
| 6. Indoor environmental quality | 28% |
| 7. Other (please specify) | EE in buildings; Passive design; Passive and active EE strategies; Resilient urban design; Politics; Lighting; Health and wellbeing |

**Research questions**

Rating scale: 1-lowest level of relevance; 5-highest level of relevance

| 1 | 2 | 3 | 4 | 5 |
|---|---|---|---|---|
| Lowest level of relevance/impact | Low level of relevance/impact | Middle level of relevance/impact | High level of relevance/impact | Highest level of relevance/impact |

1. What is Sustainable Design/Architecture significance in achieving Agenda 2030 Sustainable Development Goal number/Good health and well-being?

Answer:

| Sustainability | Agenda 2030 SDG nu. 3: Good Health and Well-Being | | | | |
|---|---|---|---|---|---|
| | Level of Relevance/Impact (x) | | | | |
| | 1 | 2 | 3 | 4 | 5 |
| Sustainable design/Architecture | | | 6% | 44% | 50% |

2. On a 1–5 scale, please rate the sustainable design category's relevance for building users' Health and well-being.

| 1 response = 6.25% | 5 responses = 31.25% | 9 responses = 56.25% | 13 responses = 81.25% |
|---|---|---|---|
| 2 responses = 12.5% | 6 responses = 37.5% | 10 responses = 62.5% | 14 responses = 87.5% |
| 3 responses = 18.25% | 7 responses = 43.75% | 11 responses = 68.75% | 15 responses = 93.75% |
| 4 responses = 25% | 8 responses = 50% | 12 responses = 75% | 16 responses = 100% |

| Sustainability Category (According to the LEED Rating System) | Building Users' Health and Well-Being | | | | |
|---|---|---|---|---|---|
| | Level of Relevance/Impact (x) | | | | |
| | 1 | 2 | 3 | 4 | 5 |
| 1. Location and transportation | | 6.25% | 25% | 43.75% | 25% |
| 2. Sustainable sites | | 6.25% | 18.75% | 50% | 25% |
| 3. Water efficiency | 6.25% | 12.5% | 31.25% | 18.75% | 31.25% |
| 4. Energy and atmosphere | | 6.25% | 12.5% | 56.25% | 25% |
| 5. Materials and resources | | 6.25% | 31.25% | 25% | 37.5% |
| 6. Indoor environmental quality | | | 6.25% | 25% | 68.75% |

3. What is the significance/impact of Passive Design measures on providing comfort indoors?

| Sustainable Design | Comfort Indoors | | | | |
|---|---|---|---|---|---|
| | Level of Relevance/Impact (x) | | | | |
| | 1 | 2 | 3 | 4 | 5 |
| Passive design measures | | | 18.75% | 37.5% | 43.75% |

4　Please rate the significance/impact of Comfort related aspects for building users' health and well-being within Indoor Environment.

| Comfort Category/Aspects | Building Users' Health and Well-Being | | | | |
| --- | --- | --- | --- | --- | --- |
| | Level of Relevance/Impact (x) | | | | |
| | 1 | 2 | 3 | 4 | 5 |
| 1. Thermal comfort | | 6.25% | | 43.75% | 50% |
| 2. Air quality | | 6.25% | 6.25% | 31.25% | 56.25% |
| 3. Visual Comfort | | 6.25% | 12.5% | 18.75% | 62.5% |
| 4. Acoustic Comfort | | | 18.75% | 37.5% | 43.75% |
| 5. Biophilic Aspects of Comfort | 6.25% | | 18.75% | 25% | 37.5% |
| 6. Psychological/Social Aspects of Comfort | 6.25% | | 18.75% | 12.5% | 62.50% |

## I Thermal comfort

(a)　Please rate the following Thermal comfort Assessment Passive Design Criteria & Indicators& Measures' relevance for enhancing building users' health and well-being.

| Comfort Assessment Category | | Building Users' Health and Well-Being | | | | |
| --- | --- | --- | --- | --- | --- | --- |
| I Thermal Comfort | | Level of Relevance/Impact (x) | | | | |
| Criteria | Indicators& Passive Design Measures | 1 | 2 | 3 | 4 | 5 |
| 1. Form, orientation | 1.1 Building geometry(compactness, volume) | | | | 50% | |
| | 1.2 Building Orientation | | | | 62.5% | |
| | 1.3 Rooms Orientation | | | | 62.5% | |
| 2. Passive solar heating | 2.1 Passive solar systems | | | | 56.25% | |
| | 2.2 Materialization | | | | 56.25% | |
| 3. Passive cooling | 3.1 Overheating Prevention | | | | 37.5% | |
| | 3.2 Passive cooling | | | | 50% | |
| 4. Thermal insulating | 4.1 Earth-sheltering | | | | 43.75% | |
| | 4.2 Green roofs &facades | | | | 37.5% | |
| | 4.3 Materialization | | | | 50% | |
| 5. Windshield | 5.1 Natural barriers | | | 31.25% | | |
| | 5.2 Artificial barriers | | | 50% | | |

(b)　Is there any other passive design thermal comfort assessment criteria/indicator/measure that you would add? If yes, please specify.

Answer:

## II Air quality

(a)　Please rate the following Air Quality related Passive Design assessment criteria relevance for enhancing building users' health and well-being.

| Comfort Assessment Category | | Building Users' Health and Well-Being | | | | |
| --- | --- | --- | --- | --- | --- | --- |
| **II Air Quality** | | Level of Relevance/Impact (x) | | | | |
| **Criteria** | **Indicators& Passive Design Measures** | 1 | 2 | 3 | 4 | 5 |
| **1. Air Cleaning** | 1.1 Vegetation | | | | 50% | |
| | 1.2 Water features | | | | 50% | |
| **2. Providing healthy Air exchange rate** | 2.1 Natural ventilation | | | | | 68.75% |
| | 2.2 Breathing walls | | | | | 50% |
| **3. EMF reduction** | 3. Increased distance from EMF sources | | | | 43.75% | |
| **4. Avoidance of geopathic zones** | 4.1 Increased distance from sources of radon | | | | 50% | |
| **5. Materialization** | 5.1 Hygroscopic materials | | | | | 43.75% |
| | 5.2 Non-toxic materials | | | | | 68.75% |

(b)　Is there any other passive design air quality assessment criteria/indicator/measure that you would add to the above-mentioned list? If yes, please specify.

Answer:

## III Visual Comfort

(a)　Please rate the following Visual Comfort related Passive Design assessment criteria/indicator/measure' relevance for enhancing building users' health and well-being.

| Comfort Assessment Category | | Building Users' Health and Well-Being | | | | |
| --- | --- | --- | --- | --- | --- | --- |
| **III Visual Comfort** | | Level of Relevance/Impact (x) | | | | |
| **Criteria** | **Indicators& Passive Design Measures** | 1 | 2 | 3 | 4 | 5 |
| **1. Daylighting/windows** | **1.Windows size** (Minimum vertical windows size equal to 15% of the floor area/daylighting factor of 2%) | | | | | 68.75% |
| | **1.2 Windows layout** | | | | | 68.75% |
| **2. Avoidance of glare** | **2.1 Shading devices** | | | | | 37.5% |
| | **2.2 Materialization** | | | 43.75% | | |
| **3. Visually stimulating design** | **3.1 Activity/color/illuminance ratio** e.g., Cold colors/higher illuminance for intellectual activities; warm colors/lower illuminance for physical activity | | | | | 37.5% |

(b)　Is there any other passive design Visual Comfort assessment criteria/indicator/measure that you would add to the abovementioned list? If yes, please specify.

Answer:

## IV Acoustic Comfort

(a)　Please rate the following Acoustic Comfort related Passive Design assessment criteria/indicator/measure' relevance for enhancing building users' health and well-being.

| Comfort Assessment Category | | Building Users' Health and Well-Being | | | | |
|---|---|---|---|---|---|---|
| **IV Acoustic Comfort** | | Level of Relevance/Impact (x) | | | | |
| **Criteria** | **Indicators& Passive Design Measures** | 1 | 2 | 3 | 4 | 5 |
| **1. Noise screening** | 1.1 Acoustic barriers | | | | | 43.75% |
| | 1.2 Sound absorbers | | | | 43.75% | |
| **2. Noise masking** | 2.1 Vegetation | | | | 43.75% | |
| | 2.2 Water features | | | | 43.75% | |
| **3. Soundproofing** | 3.1 Materialization | | | | | 43.75% |

(b) Is there any other passive design Acoustic Comfort assessment criteria/indicator/measure that you would add to the abovementioned list? If yes, please specify.

Answer: Other sound distractions (e.g., presence of music, natural sounds) to reduce background noise

## V Biophilic aspects of Comfort

(a) Please rate the following Biophilic Comfort related Passive Design assessment criteria/indicators/measures' relevance for enhancing building users' health and well-being.

| Comfort Assessment Category | | Building Users' Health and Well-Being | | | | |
|---|---|---|---|---|---|---|
| **V Biophilic Aspects of Comfort** | | Level of Relevance/Impact (x) | | | | |
| **Criteria** | **Indicators& Passive Design Measures** | 1 | 2 | 3 | 4 | 5 |
| **1. Nature views** | **1.1 Windows** | | | | | 81.25% |
| | **1.2 Artwork** | | | | 37.5% | |
| **2. Access to nature** | **2.1 Artificial elements** (terraces, atriums) | | | | | 56.25% |
| | **2.2 Natural elements** (e.g., gardens, courtyards) | | | | | 81.25% |
| **3. Introducing natural elements indoors** | **3.1 Retaining existing natural elements** (rock, tree) | | | | 37.5% | |
| | **3.2 Natural features** (water, greenery) | | | | 43.75% | |
| | **3.3 Four fundamental elements** (air, fire, water, earth) | | | 37.5% | | |
| **4. Biomimicry-imitation of natural forms** | **4.1 Shapes** (curved shapes, segments of space, architectural details) | | | | 31.25% | |
| **5. Materialization** | **5.1 Natural materials** | | | | | 62.5% |

(b) Is there any other passive design biophilic comfort assessment criteria/indicator/measure that you would add? If yes, please specify.

Answer:

## VI Psychological/Social aspects of Comfort

(a) Please rate the following Psychological/Social Comfort related Passive Design assessment criteria/indicators/measures' rele-vance for enhancing building users' health and well-being.

| Comfort Assessment Category | | Building Users' Health and Well-Being | | | | |
|---|---|---|---|---|---|---|
| **VI Psychological/Social Aspects of Comfort** | | **Level of Relevance/Impact (x)** | | | | |
| **Criteria** | **Indicators& Passive Design Measures** | **1** | **2** | **3** | **4** | **5** |
| **1. Constant and controlled change** | **1.1 Physiological** (visual, thermal, aerial, acoustic) | | | 37.5% | 37.5% | |
| **2.Visual aspects** | **2.1 Views** (nature, artwork) | | | | | 62.5% |
| | **2.2 Color** (cold colors/warm climates, intellectual activities; warm colors/ cold climates, physical activities) | | | | 43.75% | |
| | **2.3 Form** (human-scale, soft, curved shapes supporting activity, rectangular in favor of intellectual processes) | | | | 50% | |
| **3. Bonding building with place and time** | **3.1 'Rooting' building into the ground** (earth-sheltering, attached vegetation) | | | | 43.75% | |
| | **3.2 Materialization** (local, autochthone materials) | | | | 43.75% | |
| **4.Adaptability and flexibility** | **4.1 Physical** (space adaptable in layout and size) | | | | | 56.25% |
| | **4.2 Functional** (space adaptable in use) | | | | 50% | |
| **5. Safety and Accessibility** | **5.1 Physical** (access for disabled) | | | | | 56.25% |
| | **5.2 Social** (Psychological) / wayfinding, human scale, hospitable, familiar, 'domestic' atmosphere | | | | | 56.25% |
| **6. Social Support** | **6.1 Physical** (rooms adaptable to different size groups, furniture layout/ round tables instead of rectangular, seating 'in circle' instead of in rows) | | | | 43.75% | |
| **7. Materialization** | **7.1 Tactility of materials** (warm, natural materials, materials pleasant to walk on (sand, gravel, pebbles) | | | | 37.5% | 37.5% |

(b)    Is there any other passive design psychological/social comfort assessment criteria/indicator/measure that you would add? If yes, please specify.

Answer:

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
