# Peer review of "Toward Regenerative Sustainability: A Passive Design Comfort Assessment Method of Indoor Environment"

_sustainability, doi:10.3390/su15010840_

Round 1
Reviewer 1 Report
This was an engaging and stimulating read; I appreciate the wide scope you employ.
Author Response
Dear Reviewer,
Thank you for the compliments.
Kind regards,
Authors
Reviewer 2 Report
Congratulations on writing an excellent research paper.
However, I have few remarks. You conducted focus groups of experts (see lines 293-302) so I suggest writing how many experts participated, when did you conducted focus groups and what was the procedure of selecting experts. Secondly, please consider shortening your paper a little bit.
Author Response
Dear Reviewer,
Thank you for the compliments and the remarks.
Firstly, we amended the paper with explanations and further specifications of the applied method-focus group, involving 16 experts contacted during the last month (October). The experts were chosen according to the involvement in the key paper-related regenerative design (sustainability) topics: bioclimatic (sustainable) architecture, indoor environment, comfort, health, and well-being for more than 10 years. The majority are University professors (8), than Architects-designers (6) having awarded competition projects addressing sustainability issues, and, finally, scientific researchers (2).
Secondly, we applied some text reductions in reference to the chapter of thermal comfort. However, due to the demand of introducing more text, the final number of pages have remained the same. Our opinion is that the paper should not be shortened more in order not to lose a valuable content.
Kind regards,
Authors
Reviewer 3 Report
The topic is interesting and the paper comprises a comprehensive literature review. However, the paper shall be presented with a clear research gap, aim and objectives. The content shall be selected as fit into the selected objectives for the journal paper.
Some specific comments are attached herewith.

Author Response
Dear Reviewer,
Thank you for the useful feedback.
Firstly, the introduction section is presented (amended) with the research gap clarified: the insufficient presence of regenerative design/assessment methodology comprising diverse, but intertwined key topics: bioclimatic/passive design, biophilic design, and comfort-related aspects, all in reference to the indoor environment. The concept of salutogenesis reflects the main research approach: addressing crucial comfort-related architectural factors having impact on physical, mental and social well-being, in difference to primarily physiological factors present in the recent sustainable design/assessment methodologies.
Secondly, according to your suggestions, we introduced a new chapter two: “Literature Review” encompassing the following sub-headings: regenerative sustainability, salutogenesis, bioclimatic architecture and passive design, biophilic design, and comfort, with more clearly established symbiotic relations, interconnections and interdependencies, all relevant for enhancing building users’ health and well-being.
The scope of the paper is now stated more clearly. Only comfort categories/criteria/indicators are to some extent adopted from the doctoral thesis. Regenerative sustainability and biophilic design framework is newly introduced in the paper, as well as the comparative analysis and the focus group method. The chosen sustainability rating systems are considered as most popular and widely spread worldwide. Also, the focus group method is now clearly explained and more specifically presented.
Furthermore, the introduced Passive design comfort assessment model is a conceptual model derived from the literature review, for which we added explanation in the chapter 3 and 4.
Finally, the main objective of the paper is enhancing regenerative design comfort related methodology by the proposed passive design model addressing highly relevant aspects for enhancing health and well-being of building users in reference to the indoor environment, lastly, contributing to fulfilling Agenda 2030’s sustainable development goal number 3: good health and well-being.
Kind regards,
Authors
Round 2
Reviewer 3 Report
Thanks for the improvements made for clarity.
Introduce table 1, table 14 before they appear in the text. Check this for all the tables and figures.
A research gap has been added to the conclusion section, but it is more appropriate in the introduction section.
Results and discussion can be presented in a separate section rather than included in the conclusion section.
Author Response
Dear Reviewer,
We have introduced the table 1 (line 332) and table 14 (line 809) in the previous manuscript. Now, they appear in line 336 and 803. Also, the other tables have been checked.
A research gap was mentioned in the Introduction section (paragraph 5) instead of the Conclusions section, as you suggested.
Furthermore, Results and discussion have been presented as a separate section (number 5), followed by the section 6. Conclusions.
Kind regards,
Authors